# An Ultra-low Power RNN Classifier for Always-On Voice Wake-Up Detection Robust to Real-World Scenarios

## ABSTRACT

We present in this paper an ultra-low power (ULP) Recurrent Neural Network (RNN) based classifier for an always-on voice Wake-Up Sensor (WUS) with performances suitable for real-world applications. The purpose of our sensor is to bring down by at least a factor 100 the power consumption in background noise of always-on speech processing algorithms such as Automatic Speech Recognition, Keyword Spotting, Speaker Verification, etc. Unlike the other published approaches, we designed our wake-up sensor to be robust to unseen real-world noises for realistic levels of speech and noise by carefully designing the dataset and the loss function. We also specifically trained it to mark only the speech start rather than adopting a traditional Voice Activity Detection (VAD) approach. We achieve less than 3% No Trigger Rate (NTR) for a duty cycle less than 1% in challenging background noises pooled. We demonstrate the superiority of RNNs on this task compared to the other tested approaches, with an estimated power consumption of 30 nW in 65nm CMOS and a minimal memory footprint of 0.52 kB.

## CCS CONCEPTS

• Computing methodologies~Artificial intelligence~Natural language processing~Speech recognition • Computer systems organization~Embedded and cyber-physical systems~Embedded systems~Embedded hardware

## KEYWORDS

Voice Activity Detection (VAD), Wake-Up Sensor, Recurrent Neural Network (RNN), Gated Recurrent Unit (GRU), Machine Learning, Internet of Things (IoT), Ultra Low Power, Approximate Quantization, ASIC

## 1 Introduction

Specific applications of speech recognition algorithms such as Automatic Speech Recognition, Keyword Spotting or Speaker Verification, embedded on mobile platforms or Internet of Things (IoT) devices, are always listening to process incoming speech. When power consumption is key, it is important to avoid wasting energy by processing background noise.

For that purpose, a few ULP always-on VADs [2,14,16,20,23] have been proposed to wake the downstream speech recognition algorithms up only in the presence of speech. Their power consumption ranges between 142 nW and 22.3 µW. This is very small even if a speech recognition engine consumes 10mW.

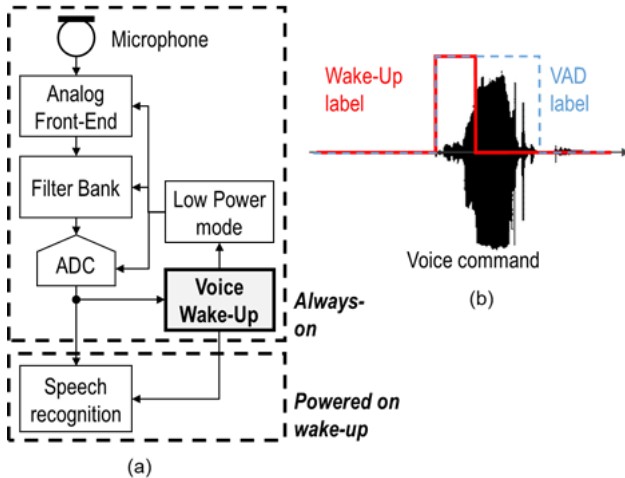

**Figure 1: (a) System diagram of a Voice WUS enabled speech recognition. (b) Example of voice command with Wake-Up and VAD training labels**

However, there is a cost on the overall system performances. A state-of-the-art small footprint keyword spotting achieves 2.85% False Rejection Rate for 1 False Accept per hour at 5 dB Signal-to-Noise Ratio (SNR) [1]. Typical performances of the published ULP VADs are around 90% speech hit rate, 90% noise hit rate with 10 dB SNR. The extra latency also needs to be added to that cost. It means that the wake-up performances need to increase significantly if these circuits are to be usable for consumer devices and worth the power savings.

In terms of algorithm, the ULP VADs all extract a short-term spectrum of the incoming audio signal on a bandwidth lower than 8 kHz, either using a bank of bandpass filters or a low resolution Fast Fourier Transform (FFT). In [2,13], the VAD algorithm relies on a small decision-tree classifier sporadically retrained online by an off-chip power hungry statistical VAD similar to [17] when a change of noise environment is detected. It makes the system robust to unseen environments but this dual algorithm adds complexity compared to a single classifier, and it still displays insufficient performances with hit rates lower than 90% in 12 dB noises. In [20], authors also adopted a decision-tree classifier but could not achieve robustness to unseen noise with no online retraining.

Multi-layer neural networks have also been used to do a per-frame classification, following a detection of the modulation frequency [14], or directly on the filter bank output [16]. Authors in [23] concatenated 3 frames separated by 30 ms and fed it to a binarized Multi-Layer Perceptron (MLP) to make the VAD decision using temporal information. The network achieved an accuracy similar to

other published work, but by training and evaluating on the same noise.

All the previous implementations are VADs. Voice activity detection is a key component of many speech recognition applications, but its purpose is to provide a label for speech and noise in order to apply different processing on speech and noise segments. We chose to train our system specifically for a simpler voice wake-up task instead of VAD. Our goal is only to mark the starting point of a speech segment and signal that there is new speech data available to downstream speech recognition algorithms. Figure 1 (b) shows the difference in labelling between VAD and WUS. The idea is that a simpler task should need a simpler detector.

As for the nature of the detector, we looked at VAD algorithms from before the neural network era, exploiting the statistical and sequential properties of the speech, such as the Sohn VAD [11]. It calculates the input signal FFT and estimates the noisy speech statistics from the noise statistics per bin extracted from a noise only segment. A Markov model enables the smoothing of the per-frame decision. Another instance is the Long-term Spectral Divergence VAD [17], the SNR per FFT bin is estimated through the divergence of short term and long term variations in energy. These two techniques do not rely on any prior knowledge of the spectral distribution of speech, except for the parameter settings. Their success indicate that a classifier can extract a lot of information by examining the noisy speech spectrally but also as a sequence.

This idea makes the RNN a prime candidate for the voice wake-up task because of its ability to model arbitrary length sequences. High performance VAD algorithms [7,8] have used large Long Short Term Memory (LSTM) RNN successfully for ASR in challenging noise conditions.

Furthermore, the recursive nature of RNN is another advantage for efficient hardware implementation. We only need to store and update the hidden state from one cycle to the other, contrary to the more recent temporal convolution (TCN) topology used in [4] that has a small number of weights but needs the storage of all the frames in the analysis window.

In this paper, we first present in Section 2 the overall system and the feature generation. We then describe in Section 3 the design of the RNN-based classifier. Section 4 gives the training methodology, the data and the results. And we provide details of an implementation as a digital circuit in Section 5.

## 2  System Architecture

Figure 1 (a) shows a top-level diagram of the system we consider in this paper for implementing an ULP always-on voice WUS. The microphone output is processed in the analogue domain to produce a digitized short-term spectrum of the audio signal at a specific rate. A classifier analyzes this spectrogram to detect whether speech is starting. If this is the case, the sensor wakes up the speech recognition engine and may also set the always-on front-end to a higher performance mode.

Figure 2 (a) shows a more detailed view of the proposed WUS. The microphone captures the sound and converts it to a voltage with a given sensitivity. A Low Noise Amplifier (LNA) amplifies that voltage with a gain such that 0 dBV corresponds to 110 dB SPL. A normal conversation level at 1 m corresponds to 65 dB SPL [10]. If we consider a 20 dB margin with the analogue front-end noise, we then need a dynamic range of 65 dB.

To reduce the required dynamic range and save power, we add an Automatic Gain Control (AGC) block that applies a variable gain after the LNA. It adjusts dynamically the signal level using the variable gain output. The available gain values go from 0 to 30 dB with 6 dB steps. The objective of the AGC is to use the whole filter bank dynamic range, i.e. avoid clipping and have the signal well above the noise floor at all times. We define two thresholds: high and low. If the input signal passes the high threshold, the gain is decreased by one step to avoid clipping. If the signal is lower than the low threshold for a set period, typically 30 ms, the gain is increased by one step to maximize the use of the filter bank and ADC dynamic range. The applied variable gain is a feature used in the Wake-Up classification.

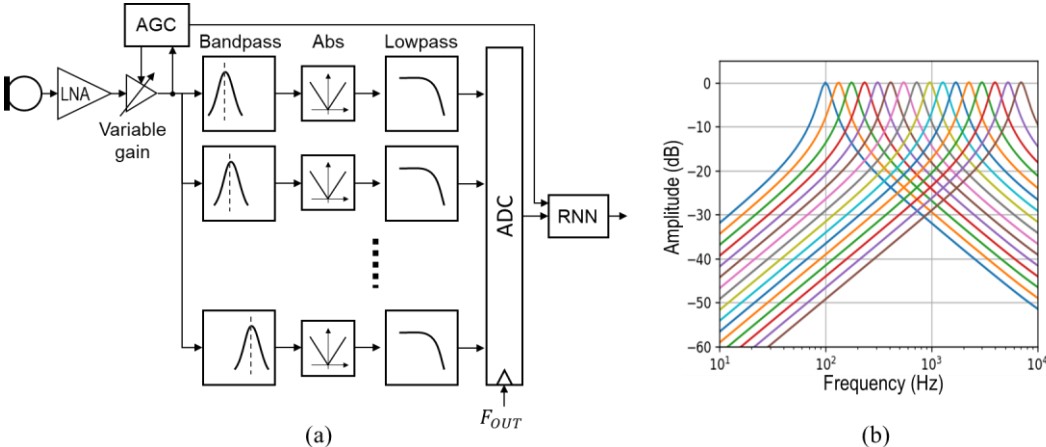

**Figure 2: (a) WUS block diagram. (b) Magnitude response of the 16 bandpass filters**

A bank of 16 first order bandpass filters extracts the signal short term spectrum. Their central frequency ranges from 100 to 7,000 Hz

with a quality factor of four regularly spaced over a logarithmic scale (see Figure 2 (b)). We chose a wider bandwidth compared to the

4 kHz and 5 kHz in [2,14,23]. Indeed, by analyzing our data, we found that starting unvoiced phones like 's', 'sh' or 'f' were more difficult to detect due to their energy at high frequency (see Figure 5 in section 4.2). We eventually used 7 kHz as our maximum frequency as speech datasets are sampled only at 16 kHz. We may obtain marginal performance gains using an even wider bandwidth.

The output of each filter goes through an absolute value function and is averaged using a first order low-pass filter with 16 Hz cut-off frequency to estimate the per-band energy in a similar way as in [2]. The output of these filters is converted to digital at a 100 Hz rate before the classification. This rate is used in literature [16,23] and it gives enough time resolution for a 100 ms target.

## 3 RNN-based Detector

### 3.1 Topology

Our goal is to have a lightweight, real-time low latency detector that makes a wake-up/no wake-up decision based on the sequence of features seen so far, as explained in section 1. The simplest topology is an RNN followed by a single dense layer.

The two most popular versions of RNNs in the literature are LSTM [9] and Gated Recurrent Unit (GRU) [3]. They show similar performances in sequence modelling task [5] but the GRU has fewer parameters for the same hidden unit dimension (two gates instead of three), which makes it more desirable for hardware implementation.

We are also interested in evaluating a simpler version of GRU called Minimal Gated Unit [24] that has a single gate, and the simple tanh-RNN. The equations of tanh-RNN, MGU and GRU are given in Table 1.

**Table 1: Equations for the tanh, MGU and GRU RNNs. $\sigma$ is the logistic function and $\odot$ is the element-wise product**

| RNN Type | Equations |
|---|---|
| tanh | $h_t = \tanh(W_{hh}h_{t-1} + W_{hx}x_t)$ |
| MGU | $f_t = \sigma(W_{fh}h_{t-1} + W_{fx}x_t)$ |
|  | $\widetilde{h_t} = \tanh(W_{hh}[f_t \odot h_{t-1}] + W_{hx}x_t)$ |
|  | $h_t = (1 - f_t) \odot h_{t-1} + f_t \odot \widetilde{h_t}$ |
| GRU | $f_t = \sigma(W_{fh}h_{t-1} + W_{fx}x_t)$ |
|  | $r_t = \sigma(W_{rh}h_{t-1} + W_{rx}x_t)$ |
|  | $\widetilde{h_t} = \tanh(W_{hh}[r_t \odot h_{t-1}] + W_{hx}x_t)$ |
|  | $h_t = (1 - f_t) \odot h_{t-1} + f_t \odot \widetilde{h_t}$ |

We removed the biases from the original equations for two reasons: simplifying the hardware by removing these extra additions and helping the quantization by having the matrix multiply result centered on 0 instead of a learned constant. We can observe that for a fixed hidden dimension, the MGU has twice as many parameters as the tanh-RNN and three times as the GRU. We will show in section 4.2 the trade-off between the complexity of the topology and the performances.

### 3.2 Quantization

Our target for the neural network implementation is a custom digital circuit to optimize the system for power. We perform two quantization steps to convert our floating-point model to a hardware compatible fixed-point one: quantizing the weights for an optimal memory footprint and quantizing the activations for a reduced complexity in the hardware operations to save power and area.

A comprehensive study of state-of-the-art CNN and RNN quantization examples has been done in [6]. According to the authors, RNNs are more difficult to quantize compared to CNNs because the quantization error on the activations accumulates over time and leads to precision issue. Quantization on 3 values {-1, 0, +1}, i.e. ternarization, has been demonstrated on just the weights on a speech recognition task [15] and on both weights and activations on a language modelling task [21]. In our experiments, we found that 4-bit quantization with only 16 recurrent units was a good trade-off between precision loss and power/area on the MGU architecture (see Figure 6).

Our quantization methodology uses some aspects of the RNN binarization recipe presented in [12] for speech de-noising task: the use of tanh for bounding the weight excursion and the incremental quantization. We define three incremental quantization levels: level zero, one and two. The weights are always kept in floating point. Level zero is training with only floating-point data: we apply a tanh function on the weights to force them between -1 and +1, and the tanh and sigmoid from the equations in Table 1 are approximated by hard versions, i.e. piecewise linear functions, given in equation (1).

$$\begin{cases} \text{hard\_sigmoid}(x) = \begin{cases} 0, x < -2 \\ \frac{x+2}{4}, -2 \leq x \leq 2 \\ 1, x > 2 \end{cases} \\ \text{hard\_tanh}(x) = \begin{cases} -1, x < -1 \\ x, -1 \leq x \leq 1 \\ 1, x > 1 \end{cases} \end{cases} \quad (1)$$

At level one, we take the pre-trained weights from level zero, we apply a k-bit linear quantization on the weights separately on each weight matrix $W_{xy}$ defined in Table 1 during the forward step. We update the underlying floating-point weights during the back-propagation. To quantize the weights, we first obtain a range by obtaining their distribution and by calculating $\sigma$, the average of their square as in equation (2) where $N$ is the number of elements in the weight matrix and $w_{ij}$ its elements. This is the expression of the standard deviation assuming that the mean is zero. We take $\pm\theta$ as the interval, with $\theta$ the unsigned quantization of $3\sigma$ as defined in equation (2). This value of $3\sigma$ would correspond to clipping 5% of the weights if their distribution was a Gaussian centered on zero (see Figure 3).

$$\begin{cases} \sigma = \sqrt{\frac{1}{N}\sum_{i,j} w_{ij}^2} \\ \theta = \frac{round(3\sigma(2^k-1))}{2^k-1} \end{cases} \quad (2)$$

The final quantization $w_{ij,q}$ of $w_{ij}$ is given in equation (3). It corresponds to a linear signed quantization with a step of $2^{k-1} - 1$ with clipping between $\pm\theta$.

$$w_{ij,q} = \begin{cases} -\theta, w_{ij} < -\theta \\ \frac{\theta}{2^{k-1}-1} round\left(\frac{w_{ij}}{\theta}(2^{k-1}-1)\right), -\theta \le w_{ij} \le \theta \\ \theta, w_{ij} > \theta \end{cases} \quad (3)$$

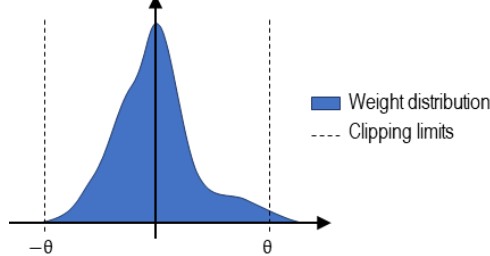

**Figure 3: Weight distribution clipping**
$\sigma$ **denotes here the average of the squared weights**

At the final quantization level, we take the weights from level one and quantize the activations as in equation (4), without retraining the weights as we found no performance improvement when doing so.

$$\begin{cases} \text{hard\_sigmoid\_qtz}(x) = \begin{cases} 0, x < -2 \\ \frac{round\left(\frac{x+2}{4}(2^k-1)\right)}{2^k-1}, -2 \le x \le 2 \\ 1, x > 2 \end{cases} \\ \text{hard\_tanh\_qtz}(x) = \begin{cases} -1, x < -1 \\ \frac{round\left(x(2^{k-1}-1)\right)}{2^{k-1}-1}, -1 \le x \le 1 \\ 1, x > 1 \end{cases} \end{cases} \quad (4)$$

## 4  Training and Performance Results

### 4.1 Training and Evaluation Data

Our goal is to train and test our WUS in a variety of noise types, speech and noise levels, and different phrases in order to estimate its robustness to real-world scenarios. Ideally, we would have used an existing benchmark for that purpose, but we found that the state-of-the-art ULP VADs [2,14,16,23] all use different source datasets with their own data augmentation strategy, limited in terms of SNRs, types of noises or number of different phrases.

We decided to create our own specific voice wake-up dataset based on publicly available datasets. We used the noise section of the MUSAN dataset [18] and speech segments from the Speech Commands dataset [22] in which we manually labelled the speech start time of 774 files. Table 2 describes how we built our training, validation and evaluation sets in terms of distribution of duration, SNR, speech and noise levels, and phrase and noise partitions. The distributions are uniform when it is a range. We designed the phrase partition to have some phonetic variability and the noise partition to feature stationary and non-stationary noises.

To create a training or test example, we take a section of noise of arbitrary length that we scale to a specified level. If it is a speech example, we mix the start of a speech command on top of the last 300 ms. We also scale it to a specified level, as illustrated on **Erreur ! Source du renvoi introuvable.** (a). The level is the RMS value of the section of speech or noise used in an example.

**Table 2: Description of the Training, Validation and Evaluation sets**

|  | Training/Validation | Evaluation |
|---|---|---|
| # Examples | 1,536/1,024 | 44,992 |
| Clip Duration | [1, 5 s] | [3, 10 s] |
| Noise Level | [-50, -30 dB] | -50; -40; -30 dB |
| Speech Level | [-46, -14 dB] | -40; -30; -20 dB |
| SNR | [9-25 dB] | 10; 20 dB |
| Phrases | Other Speech commands | "cat", "five", "no", "seven" or "wow" |
| Noises | Other MUSAN noises | "Rain_City", "Traffic", "Metro", "Birds_Park", and "White" |

### 4.2 Results

We used a custom dataset to train and evaluate the performance of our system in real-world scenarios (see in section 4.1) with the three RNNs described in section 3.1: tanh, MGU and GRU. To provide a comparison with light-weight VAD classifiers published in the literature, we evaluated three extra systems: the Sohn VAD which is end-to-end [11], and two classifiers that use the same features as in the RNNs. We used a 4-layer MLP similar to [23] with hidden dimensions of 60, 24, 11 and 1. With the first system, we classify one frame at a time to give a comparison with other single frame based VADs using decision-tree or also an MLP classifier [2,14,20]. With the second one, we take 3 frames of context as in [23] separated by 30ms before and after. We will refer to it as CMLP for Contextual MLP.

We used the same datasets and methodology to train and evaluate all the neural network based systems. We took the Adam algorithm as the optimizer with a learning rate of 0.002 for RNNs and 0.005 for MLP and CMLP, and Early Stopping on the validation loss. It took between 200 and 400 epochs to train the systems. We used a Max-Pooling loss function described in [19].

We defined three metrics to measure the performances of our system. The No Trigger Rate (NTR) is the ratio of the undetected speech examples over the total speech examples number. The False Trigger Rate (FTR) is the number of times per hour the system triggers without speech, in which case we assume that it wakes up only for 500 ms before going back to sleep mode. We target a 1% duty cycle in noise only. It corresponds to a FTR of 72 FT/hour.

We show on Figure 4 the error curves of the six systems on the entire evaluation dataset. We produce an error curve by sweeping the decision threshold and calculate the NTR and FTR for each value. In Table 3, we take the threshold corresponding to 3% NTR across all

conditions for each tested system. We show the FTR value for every noise condition and for all of them pooled.

We can see that the Sohn VAD, MLP and CMLP perform similarly at 3% NTR working point. The two extra frames of CMLP do help the detection if we compare to the single frame detection in MLP. The three RNN systems: tanh, MGU and GRU perform better by a few orders of magnitude, and we observe significant performance improvement when increasing the complexity of the RNN.

In Table 3, we can see that non-stationary noises such as "Birds Park" are a lot more challenging to detect for MLP based classifiers than for RNNs. At the opposite side, white noise is easy to detect for all systems.

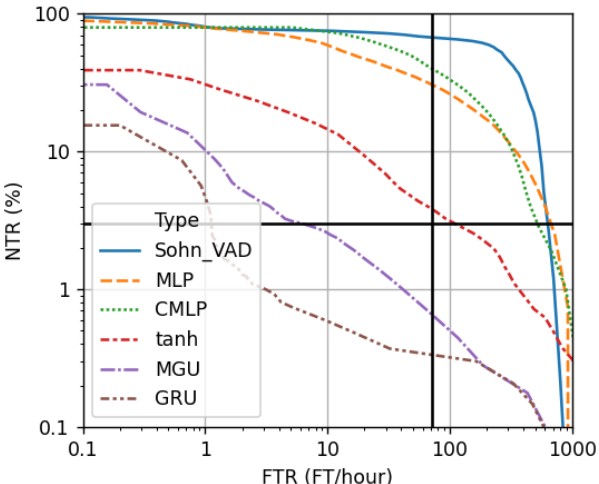

**Figure 4: Error curve for the evaluated systems across all tested conditions. The horizontal and vertical black lines show respectively the 3% NTR and 72 FT/hour working points.**

**Table 3: False Triggers per hour vs noises at 3% No Trigger Rate and the number of weights.**

| | Sohn | MLP | CMLP | tanh-RNN | MGU-RNN | GRU-RNN |
|---|---|---|---|---|---|---|
| Birds Park | 1814.1 | 1397.6 | 1425.2 | 204.1 | 19.4 | **5.7** |
| Metro | 6.2 | 798.0 | 146.5 | 69.5 | **0.0** | **0.0** |
| Rain City | 173.7 | 604.3 | 180.8 | 18.6 | **0.0** | **0.0** |
| Traffic | 68.9 | 238.7 | 443.4 | 0.2 | **0.0** | **0.0** |
| White | **0.0** | **0.0** | 31.0 | **0.0** | **0.0** | **0.0** |
| Overall | 635.9 | 684.0 | 524.7 | 110.1 | 6.2 | **1.1** |
| Num. Weights | N/A | 2,735 | 4,775 | **544** | 1,072 | 1,600 |

The last performance metric is the latency. It is the difference of time between the ground truth start and the measured start time. Figure 5 shows the system latency for different phrases. We can see that the median latency is below 100 ms in four phrases out of five. There is a strong dependence on the phonetic content of the starting phrase.

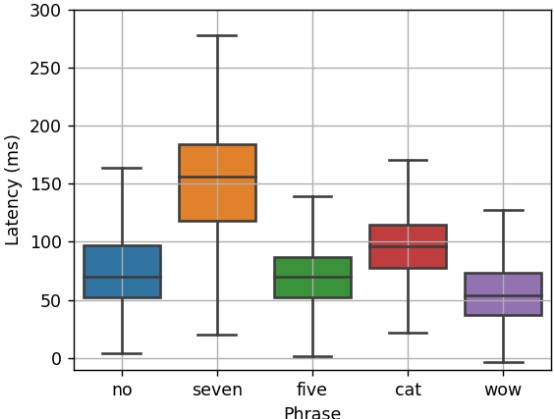

**Figure 5: Latency vs speech command with the GRU system**

Figure 6 shows the influence of quantization on detection performance. We see a degradation going from floating-point to 6, 5 and 4 bits, but the target performances are still met. The FTR at 3% NTR are respectively 16, 22 and 20 FT/hour. The degradation becomes too large when quantizing at 3-bits (189 FT/hour).

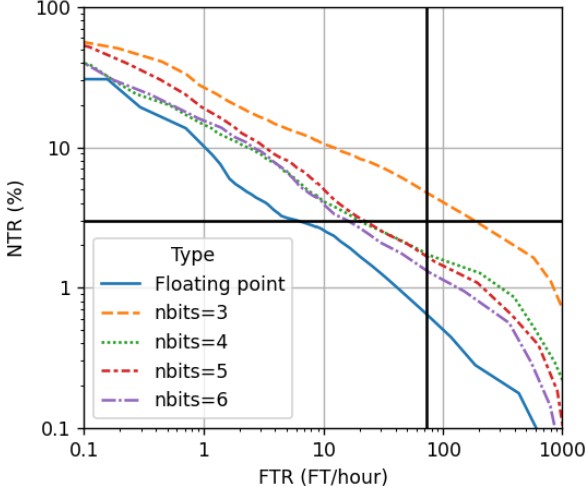

**Figure 6: Error curves for the MGU architecture in floating-point and after 3, 4, 5 or 6-bits quantization**

## 5 Digital Implementation

We tested implementing the MGU RNN topology followed by a dense layer as a digital circuit to estimate its memory footprint and power consumption. We used 16 recurrent units and 4-bits quantization for both weights and activations. Simulation results shown on Figure 6 indicate that this configuration can achieve the required performance. The total number of weights is 1,072, which takes 0.52 kB of memory. Synthesis results predict a power consumption of 45 nW under 0.9 V power supply on 65 nm CMOS process with high Vt transistors. 73% of this power is leakage. The dynamic power is small as we run only 100 inferences per second. The leakage can be further reduced by sharing a maximum of multipliers and adders. Another improvement would be to run the

calculations as sprints using a fast clock and switch off the classifier the rest of the time as in [14]. Reducing the power supply voltage from 0.9 V should also be possible, up to a certain extent.

## 6 Conclusion

We demonstrated in this work that we can train a very small-footprint and ultra-low power RNN-based detector for reducing drastically power in speech recognition applications. Our approach was to simplify the learning constraints by performing a Wake-Up instead of a VAD task. We also designed a specific dataset to test the WUS in varied and realistic unseen conditions. We have shown a large improvement over statistical VAD and single frame classification used in published ULP VADs for a smaller or comparable classifier complexity. Our quantized implementation of an MGU-RNN achieves 20 FT/hour in noise at 3% NTR in pooled conditions. Its memory footprint is 0.52 kB and we estimate a power consumption of 45 nW after digital synthesis.

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
