# OpenReview forum: "An Ultra-low Power RNN Classifier for Always-On Voice Wake- Up Detection Robust to Real-World Scenarios"
_tinyml.org/tinyML/2021/Research_Symposium — tinyML 2021 Poster_

### Official Review · AnonReviewer2 · 2021-01-27

**Overall Merit Score:** 2

**Brief Summary:**

The paper presents MGU RNN based voice activity detection digital ASIC along with its simulation results. The digital architecture is based on several techniques such as RNN, quantization, etc.


**Detailed Comments:**

The work is interesting but feels incremental. The presented ideas are mostly tried/demonstrated before, such as analog filter based feature extraction, AGC, RNN, quantization, etc.


**Paper Strengths:**

No particular strength.


**Paper Weaknesses:**

It looks like incremental works. It presents several ideas, analog filter, AGC, RNN, quantization, but they are hardly new.
Analog filter-based approach is done previously but it is not referred to properly, e.g., Sec. 2, Fig. 2(a,b).
RNN may not be the best choice. The paper didn't provide rationales for their choice. Recent works show better results with non-RNN (e.g., CNN)
The text and figures need more polishing.


**Poster (If Paper Is Rejected):**

1: Yes, ok for poster sesion to nurture work

**Reviewer Confidence:**

5: The reviewer is absolutely certain that the evaluation is correct and very familiar with the relevant literature

---

### Official Review · AnonReviewer1 · 2021-01-28

**Overall Merit Score:** 3

**Brief Summary:**

This paper does a complete stack optimization for efficient embedded "speech start" detection. It optimizes every part of the stack:

1.) The neural network algorithm and topology

2.) Its quantization and training method

3.) the used dataset and labels (speech start instead of VAD detection)

4.) The hardware mapping with power simulations

**Detailed Comments:**

* The paper is not well written. It also has very weird formatting, where e.g.
  -  text of the second column continues underneath a figure, but again starting from the first column (below fig 2)
  -  missing figure reference (Erreur ! Source du renvoi introuvable.)
  - too long intro, not well structured
* the paper claims that it has built the HW and estimates power for it, yet what is not mentioned clearly in abstract and conclusion etc. is that this is only the NN that is implemented. The power for the front-end are not reported. This results in some questions/doubts
   -  it is surprising that the proposed AGC loop is fully in the analog domain. This is good for fast adjustments, but this is likely to mess up your RNN operation/detection (as audio volume changes when the speech kicks in). Is this impact simulated / taken into account? What is the impact of the AGC step size? f the AGC update rate (30msec seems very fast...)
   - Is the front-end simulated? How? In at circuit level, or with an abstract behavioral model?
   - What accuracy / precision was expected for the front-end filters and amplifiers? Noise figure? Quantization levels? etc?
* the paper claims the MGU has more parameters then the GRU. This seems counter intuitive?
* Some confusion on the training and test set:
   - why does the test set has different noise types then the training set? Specifically, the test set does not seem to have actual noises, but only an overlay of spoken words. This is very weird, and makes me question whether actual speech start detection among noise is tested?
   - how can the evaluation and test set have so many examples if only 774 examples are manually labeled?
   - Why is the MUSAN noise database used and not the more common Aurora?
   - Will the dataset be made open source?
* the results seems weird / off:
   - one expects a concave FTR-NTR curve. The VAD Sohn (and the other methods, to a lesser extent) instead seems to be convex.
   - Table 3 shows results for different noise types, but these are (according to table 2) not in the test set?
   - Is the latency of fig 5 simulated WITH the front-end emulated?

**Paper Strengths:**

* full stack optimization, tacking all different aspects of the system to come to a globally optimal solution
* Good explanation on everything they did
* Benchmarking across a wide set of other techniques
* very low power values and good accuracy numbers achieved.

**Paper Weaknesses:**

* The paper is not well written. It also has very weird formatting
* the paper claims that it has built the HW and estimates power for it, yet what is not mentioned clearly in abstract and conclusion etc. is that this is only the NN that is implemented. The power for the front-end are not reported. This results in some questions/doubts
* the paper claims the MGU has more parameters then the GRU. This seems counter intuitive?
* Some confusion on the training and test set
* some results seems weird / off

**Poster (If Paper Is Rejected):**

1: Yes, ok for poster sesion to nurture work

**Reviewer Confidence:**

4: The reviewer is confident but not absolutely certain that the evaluation is correct

---

### Official Review · AnonReviewer3 · 2021-01-30

**Overall Merit Score:** 2

**Brief Summary:**

This paper proposed an always-on voice wake-up sensor (WUS), instead of more commonly used voice activity detector (VAD), to make the initial detection task to be simpler. A variant of GRU called minimal gated unit (MGU) has been employed, and based on experiments 16 recurrent units with 4-bit quantization are determined for the hardware design.

**Detailed Comments:**

In page 4, right column, first paragraph, there is an error on the figure number, which should be corrected.

What is the total area of the digital implementation in 65nm CMOS?

Fig. 1(a) and Fig. 2(a) shows analog modules and ADCs to precede the RNN. Shouldn’t the authors consider the power consumption of those analog modules, which I believe will far exceed the power of the digital RNN module?

The authors claimed in the abstract and conclusion that the proposed scheme is robust against unseen noise (and also criticized [23] for training and evaluating on the same noise). However, in the results section, information about the robustness against unseen noise has not been presented at all. Please elaborate how the unseen noise has been simulated and how much robustness is achieved for such unseen noise.

For Fig. 6, are both the activations and weights quantized to the specified precision? Please clarify in the text.


**Paper Strengths:**

The key idea is good, to propose a new simpler detection scheme for wake-up operations, compared to VADs, for subsequent speech recognition tasks.

**Paper Weaknesses:**

The results are somewhat weak. The digital implementation is not explained sufficiently, and the analog implementation needed for the WUS has not been reported at all.

**Poster (If Paper Is Rejected):**

1: Yes, ok for poster sesion to nurture work

**Reviewer Confidence:**

4: The reviewer is confident but not absolutely certain that the evaluation is correct

---

### Decision · Program_Chairs · 2021-02-05

**Decision:**

Accept (Poster)

**Comment:**

Based on the reviewer feedback, your paper has been accepted as a poster.

Please read the reviews carefully and make sure the concerns are addressed in your poster submission.

Accepted posters are given a 5-minute slot for an oral presentation on Friday, March 26, 2021, to pitch the main ideas of your work and to stimulate discussions. Detailed instructions will follow soon. All final posters will earn a stamp of acceptance as such: “Published as a poster at TinyML Research Symposium 2021.”